# Metformin and Hepatocellular Carcinoma Risk Reduction in Diabetic Patients with Chronic Hepatitis C: Fact or Fiction?

**DOI:** 10.3390/v15122451

**Published:** 2023-12-17

**Authors:** Marco Sacco, Davide Giuseppe Ribaldone, Giorgio Maria Saracco

**Affiliations:** Gastro-Hepatoloy Unit, Department of Medical Sciences, University of Turin, 10126 Turin, Italy; marco.sacco10@gmail.com (M.S.); davidegiuseppe.ribaldone@unito.it (D.G.R.)

**Keywords:** hepatitis C virus, chronic hepatitis C, diabetes mellitus, direct-acting antiviral agents, cirrhosis, antidiabetic therapy, interferon, metformin, hepatocellular carcinoma

## Abstract

Background: Patients with chronic hepatitis C (CHC) and concomitant type 2 diabetes mellitus (DM) show a higher risk of developing hepatocellular carcinoma (HCC). Successful antiviral therapy has reduced the incidence of post-therapy HCC, but the presence of DM still represents an unfavourable predictive factor even in cured patients. Metformin (MET) is recommended as a first-line therapy for DM, and its use is associated with a significant reduction in HCC among diabetic patients with chronic liver disease of different etiology, but very few studies specifically address this issue in patients with CHC. Aim: the aim of this review is to evaluate whether the use of MET induces a significant decrease in HCC in diabetic patients with CHC, treated or untreated with antiviral therapy. Methods: A search of PubMed, Medline, Web of Sciences and Embase was conducted for publications evaluating the role of MET in reducing the risk of HCC in patients with DM and CHC, with no language and study type restrictions up to 30 June 2023. Only studies fulfilling the following inclusion criteria were considered: (1) data on the incidence of HCC in the follow-up of diabetic patients with CHC only; (2) follow-up ≥24 months; (3) sufficient data to establish the rate of diabetic patients with CHC treated with metformin or other antidiabetic medications; and (4) data on the type of antiviral treatment and the clinical outcome. Results: Three studies met the inclusion criteria. A prospective cohort study considering only patients with DM and untreated advanced CHC, or non-responders to interferon (IFN) therapy, showed that the use of MET was associated with a significant decrease in HCC incidence, liver-related death and liver transplants. A recent retrospective study focusing on a large-scale nationwide cohort of patients with CHC in Taiwan successfully treated with IFN-based therapy stratified patients into 3 groups: non-MET users, MET users and non-diabetic patients, with 5-year cumulative rates of HCC of 10.9%, 2.6% and 3.0%, respectively, showing a significantly higher HCC risk in non-MET users compared with MET users and with non-diabetic patients, while it was not significantly different between MET users and non-diabetic patients. In a recent Italian cohort study focusing on 7007 patients with CHC treated and cured with direct-acting antiviral agents (DAAs), a combined effect of DM and MET therapy was found, showing a higher incidence of HCC in diabetic patients not taking MET compared with those without DM and those with DM taking MET. Conclusion: according to the current evidence, the use of MET should be encouraged in diabetic patients with CHC in order to reduce the risk of HCC; however, a well-designed randomized controlled trial is needed to establish the generalizability of the beneficial effects of MET in this particular subset of patients.

## 1. Introduction

Type 2 diabetes mellitus (DM) is associated with an increased risk of hepatocellular carcinoma (HCC) in patients with chronic hepatitis C (CHC) [1]. Much evidence from studies conducted in the interferon (IFN) era [2,3,4,5,6,7,8,9,10,11] showed that DM is an independent predictor of HCC, despite viral clearance. On the other hand, its role in favoring the incidence of HCC in patients with CHC successfully treated by direct-acting antiviral agents (DAAs) is still debated [12]. Recent results obtained by follow-up studies adopting DAAs [13,14,15,16,17,18,19,20,21,22,23,24,25,26,27,28,29,30,31,32] are conflicting: in many of them [13,14,15,16,17,18,19,20,21,22,23,24], DM did not emerge as a pro-carcinogenic factor through multivariate analyses, while some other studies [25,26,27,28,29,30,31,32] confirmed its detrimental role. This discrepancy may be explained by the length of follow-up, usually longer in IFN studies compared with DAAs-based studies; however, little is known about the role of concomitant medications in decreasing the neoplastic risk among hepatitis C virus (HCV)-positive diabetic patients cured with DAAs, as the great majority of follow-up studies did not report data regarding antidiabetic therapy. Metformin (MET) is one of the most commonly used medications for the therapy of DM and—according to previous studies [33,34,35,36,37,38,39,40,41,42,43,44]—its use is associated with a reduced risk of HCC, although a recent metanalysis has not confirmed this beneficial effect [45]. The presumptive protective effect of MET is based upon direct and indirect mechanisms. The direct mechanism is the decrease in insulin levels [46] while the postulated indirect mechanisms are the induction of apoptosis, the inhibition of angiogenesis, the block of cell cycles and the activation of adenosine monophosphate protein kinase [47,48,49], which is a key mediator of the tumor suppressor liver kinase B1. Regarding CHC, a recent study [50] showed that Wnt/β-catenin signaling—a well-known pathway to liver carcinogenesis—remains activated in chronic HCV-infected cells after HCV clearance with DAAs, but MET is able to reverse it.

The gold standard in establishing the role of MET in reducing the HCC risk would be a large, multicenter, randomized placebo-controlled trial, which has never been conducted. Thus, at the moment we have to rely on well-designed case-control or cohort analytic studies. 

The aim of this present review is to assess the use of MET as a protective factor of HCC in a particular subset of patients, i.e., diabetic patients with CHC treated or untreated with antiviral therapy. 

## 2. Methods

We conducted a literature search and evaluation in databases including Embase, PubMed, Medline and Web of Sciences, with no language restriction up to 30 June 2023. We included clinical trials, and case-control and cohort studies focusing on the association between the use of MET and the incidence of HCC in diabetic patients with CHC both untreated and treated with IFN and/or DAAs. The search terms used were chronic hepatitis C, diabetes mellitus, diabetes, type 2 diabetes mellitus, hepatitis C virus, HCV, direct-acting antiviral agents, interferon, insulin, metformin, cirrhosis, hepatocellular carcinoma. All the articles obtained were evaluated according to the Preferred Reporting Items for Systematic Reviews (PRISMA) guidelines [51] and accepted if they met the following criteria: (1) data on the incidence of HCC in the follow-up of diabetic patients with CHC only; (2) follow-up ≥24 months; (3) sufficient data to establish the rate of diabetic patients with CHC treated with metformin or other antidiabetic medications; (4) in patients treated with antiviral therapy, data on the type of treatment and the clinical outcome. We excluded review articles with or without metanalyses, experimental studies, duplicates, studies using the same database, case reports, updates, editorial letters and conference abstracts.

Data were independently obtained by two researchers (D.G.R. and M.S.), and discrepancies were resolved by the third reviewer (G.M.S.). The following information was collected from each study: first author, year of publication, country of origin, study and design type (randomized controlled double-blind/open label trial, observational cohort prospective/retrospective study, cross-sectional study), number of cases and controls, length of follow-up, numbers and percentage of HCC in diabetic patients treated or untreated by metformin, type of antiviral therapy. 

## 3. Results

The flow of screened papers is reported in Figure 1. Out of 7139 records identified, 6516 duplicates were removed. We excluded 564 of 623 screened reports: 186 were experimental studies, 159 meeting abstracts, 148 reviews, 28 editorials, 27 case reports and 16 letters.

Of the remaining 59 eligible papers, 21 did not evaluate the HCC risk, 13 did not include patients with CHC, 10 included patients with chronic hepatitis of various etiologies, 7 used the same database and 5 did not consider metformin.

Three studies met the inclusion criteria and were included in the qualitative analysis (Table 1).

### 3.1. Diabetic Patients with Untreated CHC or Non-Responders to Antiviral Therapy

Many studies reported the beneficial effect of MET in diabetic patients with chronic liver diseases of different etiologies [35,36,37,38,39,40,41,42,43], but only one study met the inclusion criteria of our review, specifically addressing the issue of the effects of MET on the HCC risk in untreated patients with CHC or in non-responders to antiviral therapy [34]. Nkontchou and coll. [34] followed up for about 5 years 100 biopsy-proven cirrhotic patients with HCV and DM, untreated or non-responders to IFN. Patients who obtained a sustained virological response (SVR) were excluded. Out of 100 patients, 26 (26%) were treated with MET, 17 (17%) with insulin secretagogues, 28 (28%) with insulin and 29 (29%) with diet alone. MET users and non-MET users were comparable for age, gender, BMI, DM duration, sugar control, liver enzymes, albumin and bilirubin. MET users showed higher levels of platelets compared to non-MET users (*p* = 0.04), suggesting a less advanced stage of portal hypertension. During the follow-up, 39 patients developed HCC: the 5-year incidence rate was 9.5% (95% CI: 0.0–21.4) in MET-treated patients, compared to 31.2% (95% CI: 18.7–41.8) in non-MET users (*p* = 0.001). When adjusted for several potential confounding factors (age, platelet count, BMI, alcohol intake, DM duration), the multivariate analysis showed that MET use still emerged as an independent predictive factor of decreased incidence of HCC (HR = 0.22, 95% CI 0.05–0.97, *p* = 0.046). Moreover, the authors performed a propensity score confirming the beneficial association between MET treatment and reduced HCC incidence (HR = 0.22, 95% CI 0.05–0.98, *p* = 0.047). Conversely, liver-related death or liver transplantation was less frequent among MET users compared to non-MET users (5.9% vs. 17.4%, *p* = 0.013). Interestingly, patients treated with diet alone showed a similar incidence of HCC as compared with those treated with insulin or insulin secretagogues (aHR: 1.38, 95% CI: 0.61–3.17, *p* = 0.44). The strengths of this study are the prospective design, the long-term follow-up period and the exclusion of patients with contraindications to MET. However, there are some limitations. First, the authors included a small and selected population of cirrhotic patients with DM. Second, the different groups of patients were not perfectly comparable. Third, no data on the concomitant use of statins and aspirin were reported. Last but not least, only a minority of patients (26%) were treated with MET, despite the absence of contraindications to MET in all patients, suggesting that MET users possibly had less severe DM and/or a less advanced liver disease. DM severity was not clearly defined and compared between the two groups, although the baseline median fasting glucose values and HbA1c levels were comparable. The rate of macro and microangiopathy manifestations was not reported. Met-users showed a median creatinine clearance better than non-MET users (89 mL/min vs. 82 mL/min), even though this difference was not statistically significant (*p* = 0.09). Finally, non-MET users had significantly lower levels of platelets, suggesting a more advanced liver disease. In both scenarios (i.e., more severe DM and/or more advanced liver fibrosis), the risk of developing de novo HCC is increased, making the two populations not comparable, despite adjustment with multivariate analysis.

### 3.2. Diabetic Patients with CHC Successfully Treated with Antiviral Therapy

Uncertainties remain as to the burden of DM in patients with CHC achieving SVR. Many studies conducted during the IFN era [3,4,5,6,7,8,9,10,11,12] reported a persistent unfavorable effect of DM on the HCC risk among cured patients. A recent study [52] focusing on a large-scale nationwide cohort of patients with CHC in Taiwan successfully treated with IFN-based therapy showed that diabetic patients with SVR were at an increased risk of HCC compared to non-diabetic patients. Recruitment criteria were stringent, as only patients with CHC successfully treated with IFN-based treatment were considered; patients with potentially confounding factors such as coinfection with human immunodeficiency virus or hepatitis B virus were excluded. The entire cohort was finally composed of 7249 patients followed for a median period of time ranging from 4.2 to 6.4 years. As expected, only a minority showed baseline liver cirrhosis (7.8%) due to the limited tolerance of cirrhotic patients to IFN-based therapies. Out of 7249 patients, 781 (10.8%) were diabetic and 647 of them (82.8%) were MET users. Overall, 227 (3.2%) patients developed de novo HCC, with an annual incidence of 66.9 cases per 10,000 persons-years.

When stratified according to the presence/absence of DM, patients with DM showed a higher yearly incidence of HCC compared to non-diabetics (92.4 cases per 10,000 persons-years vs. 62.6 cases per 10,000 persons-years, *p* < 0.001). However, it should be noted that diabetic patients were significantly older than non-diabetics (years: 55.9 vs. 52.3, *p* < 0.001), with more advanced liver fibrosis (36.4% vs. 25.6%, *p* < 0.001), more metabolic disorders (49.5 vs. 14.1, *p* < 0.001) and longer follow-up (years: 5.9 vs. 4.2 years, *p* < 0.001).

To establish the effect of MET use on the risk of HCC in patients with DM, the authors stratified patients into three groups: non-MET users, MET users and non-diabetic patients, with 5-year cumulative rates of HCC of 10.9%, 2.6% and 3.0%, respectively. After adjusting for several potentially confounding factors (age, gender, cirrhosis, BMI, metabolic co-morbidities, statin use), the HCC risk was significantly higher in non-MET users compared with MET users (aHR 2.84; 95% CI 1.49–5.40, *p* = 0.002) and with non-diabetic patients (aHR 2.83; 95% CI 1.57–5.08, *p* < 0.001), while it was not significantly different between MET users and non-diabetic patients (aHR 1.46; 95% CI 0.98–2.19, *p* = 0.065). These results were partially confirmed even when patients were stratified according to the presence/absence of cirrhosis: non-MET users without cirrhosis showed a higher HCC risk rate compared to MET users without cirrhosis (aHR 2.93; 95% CI 1.43–5.99, *p* < 0.001), but—surprisingly—this difference was not statistically significant in cirrhotic patients. The discrepancy was probably due to the low number of cirrhotic patients; when the authors adopted FIB-4 >3.25 as a marker of advanced fibrosis instead of cirrhosis, non-MET users (aHR 2.46; 95% CI 1.42–4.28, *p* = 0.001), but not MET users (aHR 1.24; 95% CI 0.84–1.83, *p* = 0.285), showed a significantly higher risk of HCC compared with non-diabetic patients.

Finally, to check for the potential bias of confounding factors (age, gender, cirrhosis, BMI, hypertension, dysmetabolic co-morbidities, statin use, follow-up duration), the authors applied a propensity score-matching design to assess the impact of DM/MET on HCC. HCC risk was found to be significantly higher in non-MET users (HR 5.74; 95% CI 2.80–11.79, *p* < 0.001) and in MET users (HR 1.99; 95% CI 1.12–3.53; *p* = 0.019) than in non-diabetic patients, but when only diabetic patients were considered, non-MET users still showed an increased HCC risk (HR 2.89; 95% CI 1.52–5.49, *p* = 0.001) compared to MET users. 

The strengths of this study are its homogeneous population, the high number of patients, the detailed statistical analysis trying to look for potential bias, and the long-term follow-up. However, there are some limitations related to the retrospective design of the study. First, the authors cannot exclude that the non-MET population showed contraindications to the medication (i.e., difficult-to-treat DM, impaired kidney function) usually found among patients with more advanced liver fibrosis. In particular, the mean baseline fasting glucose value in non-MET users was significantly higher than in MET users, although no difference in the levels of HbA1c and HOMA-IR was found between the two groups. However, kidney function was significantly reduced among non-MET users compared to MET-users (mean creatinine value: 1.33 ± 1.69 mg/dL vs. 0.88 ± 0.20 mg/dL, *p* = 0.002). Second, they did not establish the role of other antidiabetic medications and the interaction between MET and statins or aspirin. Third, cirrhotic patients were under-represented, making questionable the results reported in this crucial subset of patients. Fourth, the study was an Asian cohort study; thus, conclusions may not be generalized to other ethnicities.

A recent Italian study [53] reported the follow-up data of 7007 patients with CHC treated and cured with DAAs. Most of them showed baseline advanced fibrosis (4578, 65.3%), 725 out of 7007 (10.3%) had DM receiving antidiabetic medications, and 49.4% of them were given MET. After approximately 24 months of follow-up, the overall HCC incidence rate was 3.5% in 4178 patients with F3–F4 fibrosis regularly followed every 6 months. In patients with advanced fibrosis, DM was an independent predictor of de novo HCC (HR = 2.09, 95% CI 1.20–3.63, *p* = 0.009). Focusing on cirrhotic patients, a combined effect of DM and MET therapy was found, showing a higher incidence of HCC in diabetic patients not taking MET compared with those without DM (*p* = 0.0015) and those with DM taking MET (*p* = 0.018). In a multivariate analysis, the use of MET showed an independent protective association against HCC (HR = 0.32, 95% CI 0.11–0.96, *p* = 0.043). After adjustment for propensity score, the beneficial effect of MET on HCC reduction remained significant [HR = 0.24, 95% confidence interval (CI) 0.07–0.87, *p* = 0.029], as well as after restricting this analysis to patients with DM (HR = 0.22, 95% CI 0.06–0.82, *p* = 0.024). The strength of this study is based upon the relatively high number of patients included, the prospective design, the adoption of a propensity score and the consideration of several clinical and pharmacological confounders (i.e., statins, aspirin) in the multivariate analysis. However, it suffers from several limitations. First, only a minority of recruited patients were followed up for 24 months, and the duration of follow-up does not permit a reliable assessment of HCC incidence, particularly among diabetic patients who maintain the metabolic risk over time. Second, less than 50% of diabetic patients were treated with MET, and the authors cannot exclude that the remaining 50% of patients were not given MET due to more advanced DM and/or liver dysfunction. No data regarding the severity of DM were described, in particular the HbA1c levels in the two populations. Third, dose and duration of antidiabetic therapy were not reported, making questionable the real number of patients effectively treated with MET or other types of medications. Fourth, results were obtained in a prevalently Caucasian population of North Italy, and the final conclusions may not be applicable to countries with different ethnic components. In this regard, it is appropriate to compare the results of the two studies conducted on two populations with different ethnic characteristics. They are comparable regarding the number of included patients (7249 vs. 7007), male gender (53% vs. 57%), median BMI (24.6 vs. 24.7), DM rate (10.8% vs. 10.3%), but significantly differ in median age (years: 52.7 vs. 61.0), liver fibrosis severity (advanced liver fibrosis: 26.7 vs. 65.3%), MET medication in diabetic patients (82.8% vs. 49.4%), statin use (15% vs. 3.8%) and median duration of follow-up (years: 4.3 vs. 2.0). The overall HCC incidence was not comparable: Tsai and coll. [52] performed a retrospective study on the whole cohort showing a cumulative HCC incidence of 3.2%, while Valenti and coll. [53] prospectively surveilled for HCC only patients with advanced fibrosis (F3–F4, according to Metavir) as recommended by international guidelines [54], reporting an overall HCC incidence of 3.5%. The apparent similar rate of HCC incidence is puzzling, considering that the Italian population was significantly older and with more advanced liver fibrosis than the Taiwanese cohort, but the difference in follow-up length may partially explain the discrepancy. However, in a previous study conducted by the same Taiwanese group [35], about 30% of patients developing HCC after achieving SVR had fibrosis stage 1 or 2, suggesting the presence of pro-carcinogenic co-factors other than advanced liver fibrosis. 

## 4. Discussion

According to the conclusions reported by the authors of the three studies considered, the use of MET in diabetic patients with CHC determines a significant reduction in the incidence of HCC. This finding is in accordance with the vast majority of studies addressing the anti-oncogenic effect of MET in patients with DM and chronic liver disease of different etiology [35,36,37,38,39,40,41,42,43,44]. The rationale for this beneficial effect lies in its action on multiple metabolic pathways. MET activates adenine monophosphate-activated protein kinase (AMPK), which reduces hepatic gluconeogenesis, inhibits fatty acid synthesis and promotes fatty acid oxidation, reducing hepatic steatosis [47,48,49]. Moreover, AMPK blocks cMyc and p53, two pro-oncogenic factors, and nuclear factor kappa B (NF-kB) activation, and it decreases interleukin-6 (IL)-6 production, thus reducing inflammation. In addition, insulin sensitivity improvement and oxidative stress reduction are two well-known effects achieved with MET. Therefore, the pharmacologic background of MET supports the anti-oncogenic effects reported by the clinical studies. However, some caveats should be taken into consideration before drawing definite conclusions.

The first and more important caveat is the possible selection bias present in the abovementioned studies. Despite MET being considered a first-line treatment of DM, only 26% and 49.4% of diabetic patients considered in the study of Nkontchou and coll. [34] and Valenti and coll. [53], respectively, were treated with MET. In the Taiwanese cohort [52], the great majority (82.8%) of diabetic patients were given MET. What is the reason why all diabetic patients were not treated by MET? One possible explanation is related to MET-induced side effects such as nausea, vomiting, abdominal discomfort and diarrhea, but these data cannot be derived from the studies. A second possible explanation may be based upon the severity of diabetic disease; difficult-to-treat DM may need medications more powerful than MET. In this case, the HCC risk in non-MET users may be higher due to a greater likelihood of exposure to insulin and other antidiabetic treatments which may be more carcinogenic. In the French study [34], HbA1c levels were comparable between MET users and non-MET users, just as in the Taiwanese study [52]; however, according to Tsai and coll., fasting glucose was significantly higher in non-MET users than in MET users. Glycemic control in diabetic patients with liver disease is not trivial, as the risk of HCC is positively associated with poor glycometabolic control [55]. Last but not least, MET is less often prescribed in older patients or in those showing liver or renal impairment, that is, patients with more advanced liver disease who have a significantly higher risk of developing HCC compared to patients with less severe liver fibrosis. According to Nkontchou and coll. [34], patients with contraindications to MET were not included in the study, but non-MET users were older and showed a significantly lower platelet count compared with MET users, suggesting a possible selection bias based upon the diabetologists’ reluctance to prescribe MET to cirrhotic patients with more advanced liver disease. In the Taiwanese study [52], the difference of mean FIB-4 value between MET users and non-MET users was almost statistically significant (3.1 vs. 3.56, *p* = 0.053). In their multivariate analysis, non-MET users, but not MET users, with baseline advanced fibrosis (FIB-4 ≥ 3.25) showed a significantly higher risk of HCC compared to non-diabetic patients at a similar fibrosis stage; however, the number of MET users with FIB ≥ 3.25 was very low, making the conclusions questionable.

Both the Taiwanese and Italian studies used propensity score-matched cohorts to look for the potential bias of confounding risks (age, sex, BMI, DM, hypertension, statin use, follow-up duration, alcohol use, coinfection with HIV, cirrhosis), confirming the higher risk of HCC among non-MET users compared to MET users. However, recent studies [21,23,56,57,58,59] showed that the HCC risk changes according to different baseline and/or dynamic parameters in cirrhotic patients achieving SVR. In other words, not all cirrhotics are the same, and the incidence rate of HCC may vary according to baseline characteristics or changes that occurred after successful antiviral therapy.

These caveats highlight the need of launching a large, multicenter, randomized controlled trial to assess the efficacy of MET in reducing HCC incidence in diabetic patients with chronic hepatitis C, even if cured by antiviral treatment. In the meantime, however, it is necessary to remember that MET is among the cheapest and safest medications for the therapy of DM, and its use should be recommended in all diabetic patients with CHC. Recent data [60] have definitively certified the safety of MET in patients with all-cause chronic liver disease and with different stages of liver fibrosis and cirrhosis, showing safe plasma lactate and MET levels in such patients. Therefore, the optimization of the treatment of DM in patients with CHC and advanced liver disease should become a primary goal in the minds of hepatologists.

## Figures and Tables

**Figure 1 viruses-15-02451-f001:**
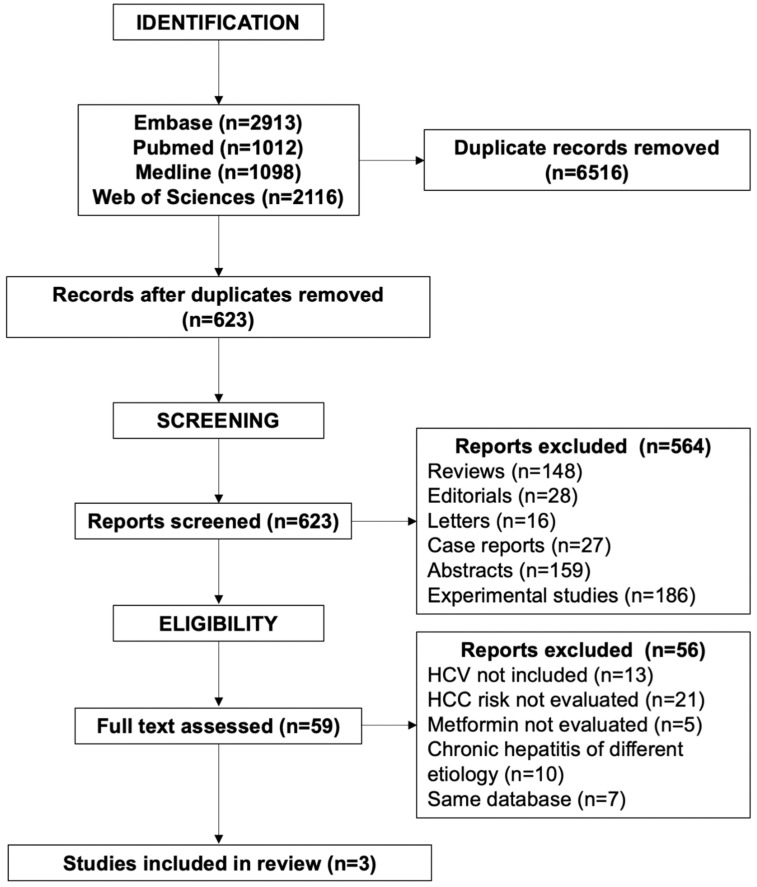
PRISMA flowchart showing the identification and selection of studies.

**Table 1 viruses-15-02451-t001:** Characteristics of studies.

Author, Year (Reference)	Nkontchou, 2011 [34]	Tsai, 2023 [52]	Valenti, 2022 [53]
Type of study	Observational	Observational	Observational
Design	Prospective	Retrospective	Retrospective
HCV-cured patients	NO	YES	YES
Diabetic patients, n	100	781	725
Cirrhosis, %	100%	10.2%	78.6%
Median follow-up, years	5.7	4.4	2.0
MET users, %	26%	82.8%	49.4%
HCC incidence MET vs. non-MET users, HR [CI]; p	0.22 [0.05–0.97]; *p* = 0.046	2.84 [1.49–5.40]; *p* = 0.002	0.24 [0.07–0.87]; *p* = 0.0029
Propensity score	YES	YES	YES

HCV, hepatitis C virus; MET, metformin; HCC, hepatocellular carcinoma; HR, hazard ratio; CI, confidence interval.

## Data Availability

Data sources can be provided by Authors upon request.

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
