# Peer review of "Metformin and Hepatocellular Carcinoma Risk Reduction in Diabetic Patients with Chronic Hepatitis C: Fact or Fiction?"

_viruses, 2023, doi:10.3390/v15122451_

Round 1

Reviewer 1 Report

Comments and Suggestions for Authors

The aim of the study was to evaluate the role of metformin in reducing the risk of HCC in patients with diabetes mellitus and chronic hepatitis C based on the literature research. However, final analysis included only three studies, all of which have important limitations. E.g. in the study by Nkontchou et al. study groups were not comparable in terms of liver cirrhosis advancement, while in the study of Valenti et al. only a minority of recruited patients was followed-up for 24 months, although it was one of the main inclusion criteria adopted by the authors of the manuscript. Also the comparison between studies cannot be performed reliably - for instance the comparative analysis of studies by Tsai et al. and Valenti et al. presented in lines 229-243 is limited by the fact that the rate of patients with advanced liver fibrosis, which is by no means the dominant factor influencing the development of HCC, was significantly higher in the study by Valenti (65.3% versus 26.7%). The small number and heterogenicity of the cited studies seriously impedes the clarity and value of the manuscript. Even though authors do critically assess the accuracy and weaknesses of cited analyses, in the discussion they state after all that the use of MET in diabetic patients with CHC determines a significant reduction in the incidence of HCC, what, in the reviewer’s opinion, is an unjustifiable conclusion with too little evidence presented to support it. Perhaps the literature search would benefit from including other key terms (such as for e.g. “liver cancer”)? Or maybe authors could also refer to other viral etiologies (HBV) to enrich the manuscript with broader analysis of the suspected hypothesis?

Comments on the Quality of English Language

The manuscript is not free of minor editing and linguistic mistakes (e.g. line 24: “associated with” instead of “associated to”; line 26: “treated with” instead of “treated by”; line 34: “current evidence” instead of “evidences”), which, however, do not seriously impede its understanding.

Reviewer 2 Report

Comments and Suggestions for Authors

Sacco et al studied the efficacy of Metformin (MET) use in DM patients to assess the incidence of developing HCC in CHC patients. The review design and reporting are well written and authors concluded that MET users have significantly reduced incidence of developing HCC. They also suggested that all DM patients with CHC should be treated with Met. Although the review is interesting, there are following concerns.

1.       Why experimental papers were excluded from the study? Experimental studies could add valuable information supporting the conclusion.  

2.       Metformin is generally used for mild cases of T2DM. In severe DM, other drugs such as glipizide, insulin therapy, etc with or without metformin are used to treat. In the text nowhere the DM patients with other drugs were taken into account and analyzed. Severity of DM with HCC also has not been addressed and analyzed. This is the major weakness of the study. It is possible only mild DM patients with MET had the benefit and reduced HCC occurrence.

3.       In the Asian study, 781 (647 MET users) and in Italian study, 725 (49% MET user) are DM patients. Definitely, there will be different stages of severity and various other drugs (such as insulin therapy) are used. Whether these severity information as subgroup can be analyzed with MET and HCC development.

4.       Line 124-125, the sentence is confusing and should be rephrased. It should be total 39 patients developed HCC of the 9.5% are MET user…..

5.       In table 1, It is confusing that two rows for Met users vs non-Met users and Non-Met-users vs Met users are used. Authors should bring all results in the same measure (even it is reversed) to present to avoid confusion.

6.       Line 234, 235, and other lines,  the reference number should be given after author names.

7.       Line 222, Even in severe DM, MET is generally continued with other drugs. Thus, this argument is not valid and authors should rewrite or remove it.

8.       Minor editing in English is required.

Comments on the Quality of English Language

Minor English editing required
